# Influence of Swine Wastewater Irrigation and Straw Return on the Accumulation of Selected Metallic Elements in Soil and Plants

Siyi Li [1], Zhen Tao [1], Yuan Liu [1], Shengshu Li [1], Rakhwe Kama [1], Chao Hu [1], Xiangyang Fan [1] and Zhongyang Li [1,2,*]

[1]  Agricultural Water and Soil Environmental Field Science Observation Research Station, Institute of Farmland Irrigation of CAAS, Xinxiang 453002, China; 82101201041@caas.cn (S.L.); 82101211615@caas.cn (Z.T.); liuyuan01@caas.cn (Y.L.); 82101202119@caas.cn (S.L.); 2020y90100008@caas.cn (R.K.); huchao@caas.cn (C.H.); fanxiangyang@caas.cn (X.F.)

[2]  National Research and Observation Station of Shangqiu Agro-Ecology System, Shangqiu 476000, China

*   Correspondence: lizhongyang@caas.cn

**Abstract:** Treated livestock wastewater reuse for irrigation and straw return in arid regions have become common practices worldwide. However, many uncertainties still exist regarding the effects of the returning straw sizes on heavy metal accumulation in soil and plants under treated livestock wastewater irrigation. In a pot experiment growing maize and soybean, large (5–10 cm), medium (1–5 cm), and small (<1 cm) sizes of wheat straw were amended to assess the changes in Cu and Zn distribution in the rhizosphere, bulk soils, and plants. Groundwater and swine wastewater were used as irrigation water resources. The results showed that irrigation with swine wastewater significantly reduced soil pH and increased the concentration of soil-available potassium. Concentrations of Cu in soil were more sensitive to swine wastewater and straw application than those of Zn in soil. Swine wastewater irrigation increased the accumulation of Cu and Zn in plants with higher concentrations of Zn, while straw return tended to inhibit this increase, especially when a small size of straw was employed. In addition to providing a reference for revealing the interaction mechanism between swine wastewater irrigation and straw return, this study proposes feasible solutions to improve the efficiency of agricultural waste recycling and realize sustainable agricultural development.

**Keywords:** livestock wastewater; straw return; soil; heavy metal





## 1. Introduction

Carbon storage has globally decreased from 188 to 108 Pg (1 Pg = $10^{15}$ g), which has been caused by climate change, lower water availability, faster mineralization, cultivation, and tillage [1]. In China, soil organic carbon loss was 2.86 Pg from the 1980s to the 2000s [2]. Soil organic matter represents the largest terrestrial organic carbon stock, and a lasting increase in carbon that is stored in the soil of agroecosystems is of increasing relevance [3,4]. Soil organic matter stems from the remains of crop residues and dead organisms, manure application, wastewater irrigation, among others [5], during which straw return and livestock wastewater irrigation are two common agricultural practices that could affect soil organic matter turnover [6].

World meat production reached 337 million tons in 2020, with China being among the three highest producers of each main meat type. For instance, China alone accounted for 38 percent of the world's pig meat in 2020 [7]. The livestock industry, which is one of the most important industries in China, has benefitted human beings in many ways such as by providing food and a livelihood to people and contributing to the economy [8]. However, livestock farming produces a high amount of wastewater containing organic compounds and heavy metals [9,10]. According to the Second National General Survey of Pollution Sources (fiscal year 2017) released by China in 2020, there were 378,800 livestock and poultry farms in China in 2017, which discharged 6,048,300 tons of chemical oxygen

demand, 75,000 tons of ammonia nitrogen, 370,000 tons of total nitrogen, and 80,400 tons of total phosphorus. The direct discharge of these livestock wastewaters could lead to environmental pollution. However, livestock wastewater can be used as an alternative to freshwater in arid and semi-arid regions in agriculture with proper treatments to reduce potential heavy metal accumulation and improve both crop growth and soil health [11,12]. Livestock wastewater irrigation improves agricultural production efficiency and soil health by constructing ecological recycling agricultural systems mingling livestock breeding and planting [13]. Straw is one of the main agricultural wastes produced by planting. In 2017, the amount of recoverable straw resources in China was 674 million tons. Straw is an organic material, and its return improves soil fertility [14]. Therefore, the combined effects of livestock wastewater irrigation and straw return on soil properties and heavy metal migration in the soil–plant system deserve further attention.

Irrigation with livestock wastewater can affect soil pH and increase the contents of organic carbon, nitrogen, phosphorus, potassium, and magnesium in soil [15], which may inhibit or promote the bioavailability, migration, and transformation of heavy metals in soil, thus affecting the accumulation of heavy metals in plants. Certain studies showed that treated wastewater irrigation decreased soil pH and increased the bioavailability of heavy metals in the soil as compared with groundwater irrigation [16,17]. Kabala et al. unraveled Cu and Zn forms in the soil under long-term (100–120 years) wastewater irrigation; they verified that the total concentrations of heavy metals in irrigated soils were elevated, especially the crystalline Fe oxide fraction, the organic matter fraction of Cu, and the exchangeable fraction of Zn [18]. In the research of the Shenyang Zhangshi Irrigation Area (SZIA) of China, wastewater irrigation resulted in the severe pollution of the SZIA soils; Cu and Zn were mainly associated with less soluble fractions [19]. It is also documented that irrigation with treated wastewater significantly increased the content of Cu in soil but had no significant effect on that of Zn [20]. In addition, long-term wastewater irrigation also led to the accumulation of heavy metals in plants [21–23].

Straw also contains a lot of nutrient elements, such as carbon, nitrogen, phosphorus, and potassium, and its decomposition process after amendment to the soil can also change the soil nutrient content and soil structure [24]. Certain studies assumed that the particulate organic matter and dissolved organic matter that had formed during the decomposition of straw would bind strongly with metals in soil through functional group interactions [25], thereby reducing bioaccumulation and the potential risks of metals in crops [26,27]. Meanwhile, certain researchers observed that the increased soil pH after straw return promoted the formation of metal hydroxide, phosphate, and other precipitates, thereby enhancing the fixation of metals in soil [28,29]. It is also demonstrated that rice straw amendment promoted plant growth and increased the content of heavy metals in pore water as well as the absorption and transfer of heavy metals to grains, but heavy metal concentrations in the plants were partially offset by the increased biomass [30]. For the combined application of wastewater with straw or its derivatives, studies have shown that applying wheat straw biochar and the compost of mixed green and table waste to the soil under wastewater irrigation can reduce the potential hazard posed by wastewater-borne heavy metals, especially Cd, Cu, and Zn, to potato crops [31]. In addition, it has been reported that the combined application of straw and pig manure to soil for 17 years led to a significant increase in soil-bioavailable Cd [32]. Straw return can directly influence the abundance and activities of earthworms and other soil organisms [33], and crop straw additions along with earthworm activities can modify soil structures and the behavior of heavy metals [34]. It is proven that wheat and chickpea straw supplements used in wastewater-irrigated soils improved microbial activity and reduced the bioavailability of toxic metals in soil [35].

Despite the myriad of studies conducted on the cumulative effects of wastewater irrigation and straw return on heavy metal migration in soil–plant systems, few studies have examined the response of heavy metal mobility and distribution to straw returning with different sizes coupled with livestock wastewater irrigation. We hypothesized that, under livestock wastewater irrigation, heavy metal transportation and accumulation in soil

and crops depends on the size of the returning straw due to the difference in the difficulty of nutrient release, and straw return may inhibit heavy metal accumulation in crops under swine wastewater irrigation. To test these hypotheses, a pot experiment was conducted to evaluate the effects of the returning straw sizes on the accumulation of heavy metals in soil and crops.

## 2. Materials and Methods

### 2.1. Site Description and Preperation of Materials

The pot experiment was conducted in a vinyl tunnel at the Agricultural Water and Soil Environmental Field Science Observation Research Station of the Chinese Academy of Agricultural Sciences at Xinxiang (35.27° N, 113.93° E). Topsoil (sandy loam) of 20 cm was collected from the farmland in the suburb of Xinxiang, and it was crushed and sieved (2 mm) before being used to fill the pots. The soil properties were as follows: pH 8.56 (water–soil ratio of 5:1), electrical conductivity (EC) (water–soil ratio of 5:1) 0.287 mScm$^{-1}$, organic matter (OM) 20.25 g·kg$^{-1}$, available potassium (AK) 107.51 mg·kg$^{-1}$, available phosphorus (AP) 25.98 mg·kg$^{-1}$, copper (Cu) 22.94 mg·kg$^{-1}$, and zinc (Zn) 39.14 mg·kg$^{-1}$. Groundwater (GW) and swine wastewater (SW) were used as irrigation water resources. The groundwater was drawn from a depth of 4.5 m beneath the ground level at the experimental station. The swine wastewater was the biogas slurry fetched from the fermented anaerobic fermentation tank of an intensive pig farm near the station. The fermentation process of swine wastewater lasted about 30 days before it was used. To conform to the standard for irrigation water quality (GB 5084-2021) [36], we diluted swine wastewater 40 times using deionized water to minimize the disturbance of water quality by the dilution process. The basic properties of GW and SW before dilution are presented in Table 1. The wheat straw was also taken from the suburb of Xinxiang and was prepared in three sizes as follows: large (5–10 cm), medium (1–5 cm), or small (<1 cm). The treatment without straw was taken as the control (CK) in this experiment. The chemical properties of the straw were as follows: total carbon (TC) 452.90 g·kg$^{-1}$, total nitrogen (TN) 6.83 g·kg$^{-1}$, total potassium (TK) 14.11 g·kg$^{-1}$, total phosphorus (TP) 1.30 g·kg$^{-1}$, copper (Cu) 2.46 mg·kg$^{-1}$, and zinc (Zn) 5.75 mg·kg$^{-1}$.

**Table 1.** Basic properties of groundwater (GW) and swine wastewater (SW). The symbol "-" refers to a property that is below the detection limit.

| Water | pH | EC mScm$^{-1}$ | COD mg·L$^{-1}$ | N mg·L$^{-1}$ | P mg·L$^{-1}$ | Cu mg·L$^{-1}$ | Zn mg·L$^{-1}$ | K mg·L$^{-1}$ | Na mg·L$^{-1}$ | Cd mg·L$^{-1}$ | Pb mg·L$^{-1}$ | Ca mg·L$^{-1}$ | Mg mg·L$^{-1}$ | Cr mg·L$^{-1}$ | Mn mg·L$^{-1}$ | Ni mg·L$^{-1}$ |
|---|---|---|---|---|---|---|---|---|---|---|---|---|---|---|---|---|
| GW | 7.43 | 0.687 | 1.75 | 0.43 | 0.04 | 0.02 | 0.02 | 1.70 | 1.50 | - | - | 34.84 | 9.28 | - | 0.01 | - |
| SW | 8.78 | 7.757 | 7818.00 | 1807.50 | 219.17 | 4.96 | 12.62 | 1022.00 | 690.00 | - | 0.02 | 74.31 | 14.18 | 0.14 | 1.45 | 0.27 |

### 2.2. Plant Cultivation

A total of 80 g (7.2 t/ha) of straw of a specific size according to the local production of straw and 30 g of compound fertilizer (N-P$_2$O$_5$-K$_2$O 15:15:15, nutrient ≥ 45%) were applied to each pot (a top diameter of 45 cm, a bottom diameter of 33.5 cm, and a height of 39 cm); each pot was filled with 30 kg of soil. Four seeds of maize (*Zea mays* cv. Zhengdan958) or soybean (*Glycine max* cv. Zhonghuang13) were sown in each pot. After two weeks of germination, the seedlings per pot were reduced to 1. Before thinning, all the treatments were irrigated with groundwater. After thinning, the pots were irrigated with GW or SW according to the treatment design, and the irrigation amount was the same across all the treatments. The soil moisture of the pots was maintained at 60–70% of the water-holding capacity. The pots were irrigated every 3–4 days. A total of 1 L of water was measured using a beaker with a scale and then evenly poured into each pot each time. Each treatment was replicated three times. The pot experiment was conducted from 28 June 2021 to 9 October 2021 for soybeans. The maize seeds were initially planted on the same day as the soybeans, but the final planting time was 28 July because all the soil preparation and

seed plantation procedures were rearranged after the destructive effects of extremely heavy rainfall on 21 July 2021. The maize experiment was stopped on 4 November due to a sudden drop in temperature, which affected the development of its fruits.

### 2.3. Soil and Plant Sampling

At the maturity stage, bulk soils of 0–10 cm were collected from 3 random points higher than 10 cm away from the stem with a soil auger, and then the maize or soybean plants, including the whole root system, were carefully taken out of the pots. Soil samples shaken off the root system were collected as the rhizosphere. The soil samples for each treatment were divided into two subsamples, during which the first was stored at 4 °C for the determination of inorganic nitrogen and the second was air-dried for analysis of other properties. The plant samples were divided into classifications of roots, shoots, and fruits for maize and into classifications of roots, stems, leaves, pods, and fruits for soybeans. They were washed, oven-dried at 65 °C, ground, and stored in sealed plastic bags for further analysis. The maize grains were not enough for the experimental analysis, so the maize fruits discussed in this experiment were the those of the whole ears of maize.

### 2.4. Soil and Plant Analysis

Soil suspension with a soil-to-water ratio of 1:5 was used to determine soil pH (Orion Star A211, Thermo Fisher Scientific, Waltham, MA, USA. The potassium dichromate oxidation method was used to determine soil organic matter (OM) by an ultraviolet spectrophotometer (L9, INESA, Shanghai, China). Soil-available potassium (AK) was extracted by $NH_4OAc$ and measured by a flame photometer (FP640, INESA, Shanghai, China). Soil-available phosphorus (AP) was extracted by $NaHCO_3$ and determined by an ultraviolet spectrophotometer (L9, INESA, Shanghai, China). Nitrate-nitrogen and ammonium-nitrogen were extracted by $CaCl_2$ and determined by a flow analyzer (Auto Analyzer 3, Bran Luebbe, Norderstedt, Germany). For the determination of heavy metals in soil and plants, 0.3 g of the samples were digested with an 8 mL mixture of $HNO_3$ and HCl in a 1:3 ratio at 80 °C until a transparent solution was obtained (USEPA Method: 3005A). The cooled mixture was then transferred to a glass volumetric bottle and made up to a final volume of 50 mL with deionized water. The atomic absorption spectrophotometer (AAS PE900H, Perkin Elmer, Boston, MA, USA) was then applied to determine the concentration of Zn and Cu in the solution.

### 2.5. Statistical Analysis

The data were analyzed using SPSS 26.0 and presented as the mean $\pm$ standard deviation. Statistical comparison of soil and plant properties between treatments was assessed by the Waller–Duncan test with the significance level at $p < 0.05$. The figures were generated using Origin Pro 2023. Redundancy analysis (RDA) was used to assess the relationships between the soil and plant Cu/Zn and other soil properties in CANOCO 5. For each RDA model, statistical predictors were identified from the summarized effects and interactive forward selection of soil properties. The statistical significance of each RDA model was assessed based on 999 Monte Carlo permutations.

The bioaccumulation factor (BF) and translocation factor (TF) were calculated for Cu and Zn as follows:

$$BF = \frac{Concentration\ of\ metal\ in\ plant\ roots}{Concentration\ of\ metal\ in\ rhizosphere} \tag{1}$$

$$TF = \frac{Concentration\ of\ metal\ in\ fruits}{Concentration\ of\ metal\ in\ roots} \tag{2}$$

## 3. Results

### 3.1. Variation in the Basic Chemical Properties of Soils

For maize, no significant differences were observed in pH, organic matter, or available potassium between GW-irrigated soils and SW-irrigated soils (Figure 1). As for available phosphorus, SW irrigation increased the content of available phosphorus in bulk soils but not in the rhizosphere as compared with GW irrigation. The straw treatment of a small size had a higher content of available phosphorus in the rhizosphere than the other straw treatments under SW irrigation. There was no significant difference in the nitrate-nitrogen content in the rhizosphere irrespective of the straw sizes or the irrigation water resources. Under GW irrigation, the contents of nitrate-nitrogen in bulk soils were at a similar level; however, under SW irrigation, all the straw treatments augmented the nitrate-nitrogen levels in the soils relative to those observed for the no-straw controls, with the strongest enhancement occurring in the straw treatment of a large size. The ammonium-nitrogen content in the rhizosphere under SW irrigation was lower than that under GW irrigation, especially in the straw treatment of a large size (Figure 1).

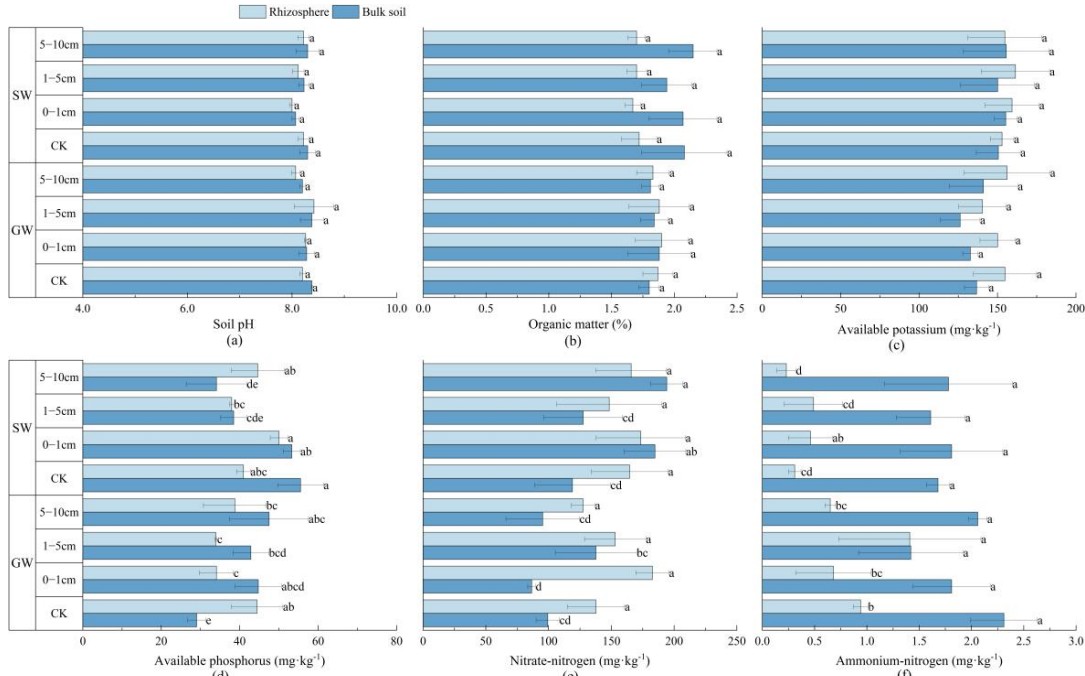

**Figure 1.** Soil pH (**a**), organic matter (**b**), available potassium (**c**), available phosphorus (**d**), nitrate-nitrogen (**e**), and ammonium-nitrogen (**f**) of the rhizosphere and bulk soil of maize under the treatments of different irrigation water resources and straw sizes. GW refers to groundwater, and SW refers to swine wastewater. Four sizes of straw (CK, 0–1 cm, 1–5 cm, and 5–10 cm) were applied to the soil. Different lowercase letters on the right of the light-blue and dark-blue columns represent significant differences in soil properties among the different treatments of rhizosphere and bulk soils, respectively at $p < 0.05$.

For soybeans, the soil pH values were higher under GW irrigation than under SW irrigation, and there were no appreciable variations in the straw return treatments (Figure 2). The contents of OM in the soil were not significantly altered with a few exceptions. SW irrigation elevated the content of available potassium in the soils compared with GW irrigation. Although straw return did not improve the content of available potassium in the soils whatever the applied irrigation water resource was, it generally increased the content of available phosphorus in the bulk soils, with the most significant enhancement being observed for the straw treatment of a large size under SW irrigation. SW irrigation boosted the content of ammonium-nitrogen in bulk soils compared with GW irrigation, while the

opposite was true for nitrate-nitrogen. Straw returns of 1–5 cm promoted the accumulation of both forms of inorganic nitrogen in the rhizosphere and bulk soils under SW irrigation.

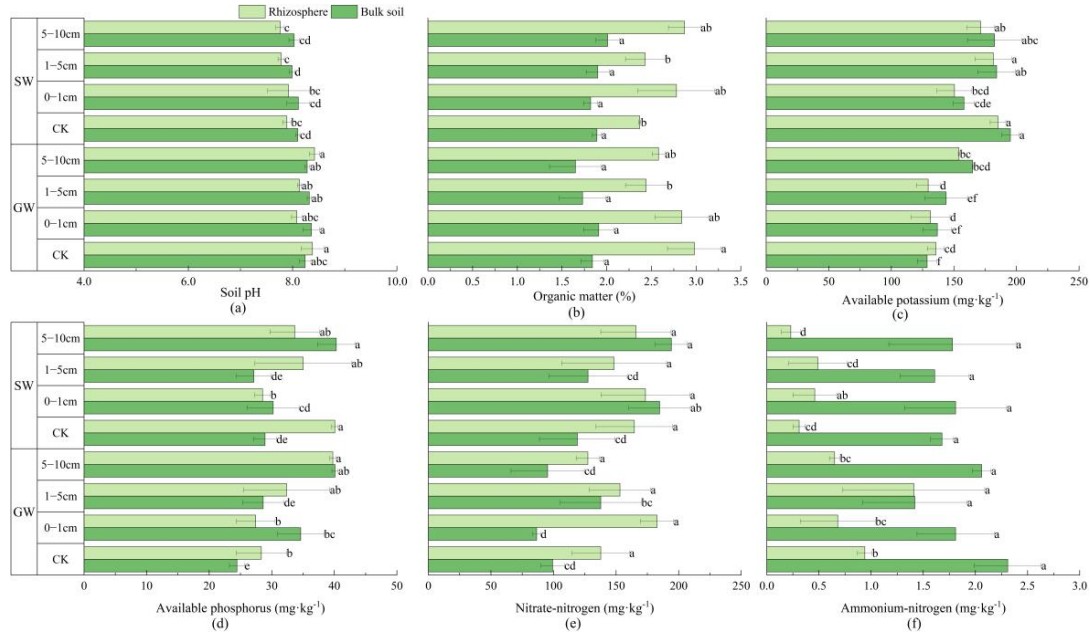

**Figure 2.** Soil pH (**a**), organic matter (**b**), available potassium (**c**), available phosphorus (**d**), nitrate-nitrogen (**e**), and ammonium-nitrogen (**f**) of the rhizosphere and bulk soil of soybeans under the treatments of different irrigation water resources and straw sizes. GW refers to groundwater, and SW refers to swine wastewater. Four sizes of straw (CK, 0–1 cm, 1–5 cm, and 5–10 cm) were applied to the soil. Different lowercase letters on the right of the light-green and dark-green columns represent significant differences in soil properties among different treatments of the rhizosphere and bulk soils, respectively, at $p < 0.05$.

### 3.2. Heavy Metals in Soils

Overall, the heavy metal contents in SW-irrigated soils were slightly higher than those in GW-irrigated soils (Figure 3). The results of two-way ANOVA showed that irrigation water resources, straw size, and their interaction had a significant effect on the Cu content in maize soils (Table 2). The Cu content in the rhizosphere of maize declined with the size of returning straw when irrigated with GW, yet it increased with the addition of straw under SW irrigation, with there being no significant influence that resulted from the straw size (Figure 4a). The Cu content in the bulk soil showed a similar trend, with a significantly lower content in the 1–5 cm straw treatment than that in the CK under GW irrigation, and there was a significantly higher content in the 0–1 cm straw treatment than that in the CK under SW irrigation. For soybeans, irrigation water resources significantly affected the Cu content in soils, while the returning straw sizes and their interaction did not have this effect (Table 3). SW irrigation showed a more positive effect on the accumulation of Cu than GW irrigation did in the rhizosphere but not in the bulk soils (Figure 4b).

**Table 2.** Two-way ANOVA results for heavy metals in maize soils. Significant differences were observed at $p < 0.05$ *, $p < 0.01$ **, and $p < 0.001$ ***.

| Soil | Source of Variation | Cu | | Zn | |
|---|---|---|---|---|---|
| | | F | p | F | p |
| | Irrigation water resource | 24.724 | <0.001 *** | 4.759 | 0.044 * |
| Rhizosphere | Straw size | 4.321 | 0.021 * | 6.891 | 0.003 ** |
| | Interaction | 7.135 | 0.003 ** | 4.803 | 0.014 * |

**Table 2.** *Cont.*

| | | | | | |
|---|---|---|---|---|---|
| Bulk soil | Irrigation water resource | 15.771 | 0.001 ** | 0.001 | 0.973 |
| | Straw size | 4.062 | 0.025 * | 2.956 | 0.064 |
| | Interaction | 17.616 | <0.001 *** | 1.011 | 0.413 |

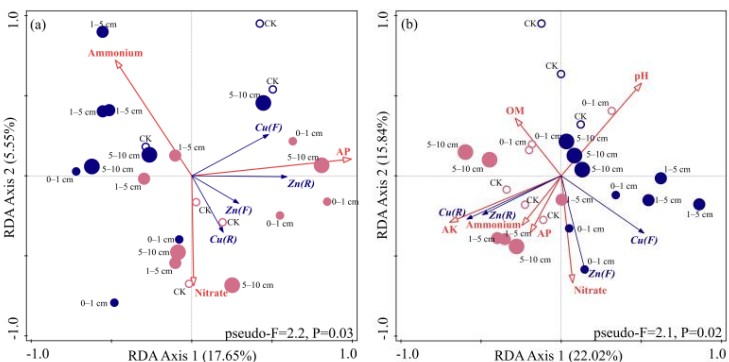

**Figure 3.** Redundancy analysis presenting the association between rhizospheric properties and Cu/Zn content in the soils and fruits of maize (**a**) and soybeans (**b**). "CK", "0–1 cm", "1–5 cm", and "5–10 cm" represent the four straw return treatments with different sizes. Cu(R) refers to the Cu content in the rhizosphere, Cu(F) refers to Cu the content in the maize or soybean fruits, Zn(R) refers to the Zn content in the rhizosphere, Zn(F) refers to the Zn content in the maize or soybean fruits, OM refers to organic matter, AP refers to available phosphorus, AK refers to available potassium, Nitrate refers to nitrate-nitrogen, and Ammonium refers to ammonium-nitrogen.

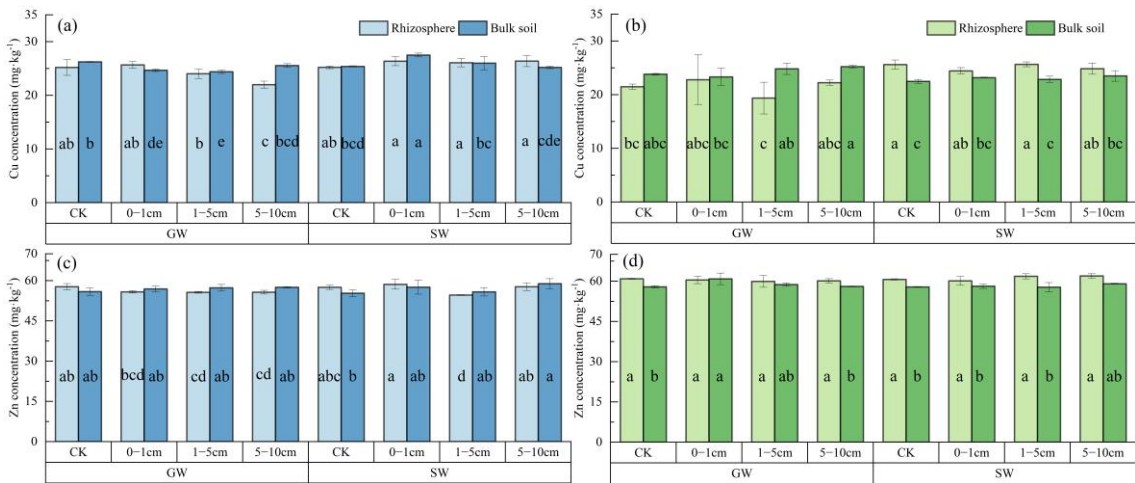

**Figure 4.** Cu concentrations in the soil of maize (**a**) and soybeans (**b**) and Zn concentrations in the soil of maize (**c**) and soybeans (**d**). "CK", "0–1 cm", "1–5 cm", and "5–10 cm" represent the four straw return treatments with different sizes. GW refers to groundwater, and SW refers to swine wastewater. Different lowercase letters on the center of the light-color and dark-color columns represent significant differences in soil properties among different treatments of the rhizosphere and bulk soils, respectively, at $p < 0.05$.

The Zn contents in the maize rhizosphere with SW irrigation were higher than those with GW irrigation (Figure 4c). The Zn content in the rhizosphere dropped after the straw return, and the drop was most obvious when 1–5 cm of straw was added under SW irrigation. There was no significant difference in the Zn content in the bulk soils among the different treatments. The Zn contents in the soybean rhizosphere were not sensitive to irrigation or straw application, but they were significantly lower in the bulk soils of the 0–1 cm straw treatment irrigated with SW than in those irrigated with GW (Figure 4d).

**Table 3.** Two-way ANOVA results for heavy metals in soybean soils. Significant differences were observed at $p < 0.05$ *, $p < 0.01$ **, and $p < 0.001$ ***.

| Soil | Source of Variation | Cu | | Zn | |
|---|---|---|---|---|---|
| | | F | *p* | F | *p* |
| Rhizosphere | Irrigation water resource | 19.618 | <0.001 *** | 2.514 | 0.132 |
| | Straw size | 0.412 | 0.747 | 0.451 | 0.720 |
| | Interaction | 1.502 | 0.252 | 1.634 | 0.221 |
| Bulk soil | Irrigation water resource | 15.081 | 0.001 ** | 2.645 | 0.123 |
| | Straw size | 2.854 | 0.070 | 2.645 | 0.085 |
| | Interaction | 1.495 | 0.254 | 3.400 | 0.044 * |

The total amount of Cu and Zn in the irrigated soil were lower than 28 mg·kg$^{-1}$ and 61 mg·kg$^{-1}$, respectively, which did not exceed the limit values for Cu and Zn in the Soil Environmental Quality Risk Control Standard for Soil Contamination of Agricultural Land (GB 15618-2018) [37].

*3.3. Heavy Metals in Plants*

For maize, the results of ANOVA (Table 4) showed that the straw size had a significant effect on the root Cu and shoot Cu and Zn, water significantly affected shoot and fruit Zn, and the interaction significantly affected shoot Zn. There were no significant differences in the Cu and Zn contents in the roots of maize except that the Cu content in the roots of maize in the straw treatment of 1–5 cm under SW irrigation was significantly enhanced compared with that in the CK with no straw under GW irrigation (Figure 5a,c). The Cu and Zn contents in maize shoots were not clearly impacted by straw return under GW irrigation, but they were slightly higher in the 0–1 cm straw treatment than in the other straw treatments. Under SW irrigation, the highest concentration of Cu in shoots was observed for the small-size straw treatment, while the highest concentration of Zn in shoots was observed for the large-size straw treatment. Straw amendment of a small size reduced both the Cu and Zn contents in maize fruits when irrigated with GW, but the straw of other sizes had no significant effect. The Cu and Zn concentrations in fruits fell due to the straw return under irrigation with SW.

**Table 4.** Two-way ANOVA results for heavy metals of maize plants. Significant differences were observed at $p < 0.05$ *, $p < 0.01$ **, and $p < 0.001$ ***.

| Parts of Plants | Source of Variation | Cu | | Zn | |
|---|---|---|---|---|---|
| | | F | *p* | F | *p* |
| Roots | Irrigation water resource | 0.124 | 0.729 | 0.023 | 0.881 |
| | Straw size | 3.718 | 0.033 * | 2.958 | 0.064 |
| | Interaction | 0.929 | 0.449 | 1.160 | 0.356 |
| Shoots | Irrigation water resource | 0.902 | 0.356 | 26.424 | <0.001 *** |
| | Straw size | 4.025 | 0.026 * | 5.122 | 0.011 * |
| | Interaction | 0.356 | 0.786 | 16.269 | <0.001 *** |
| Fruits | Irrigation water resource | 3.543 | 0.078 | 7.047 | 0.017 * |
| | Straw size | 3.141 | 0.054 | 1.372 | 0.287 |
| | Interaction | 5.750 | 0.007 ** | 0.977 | 0.428 |

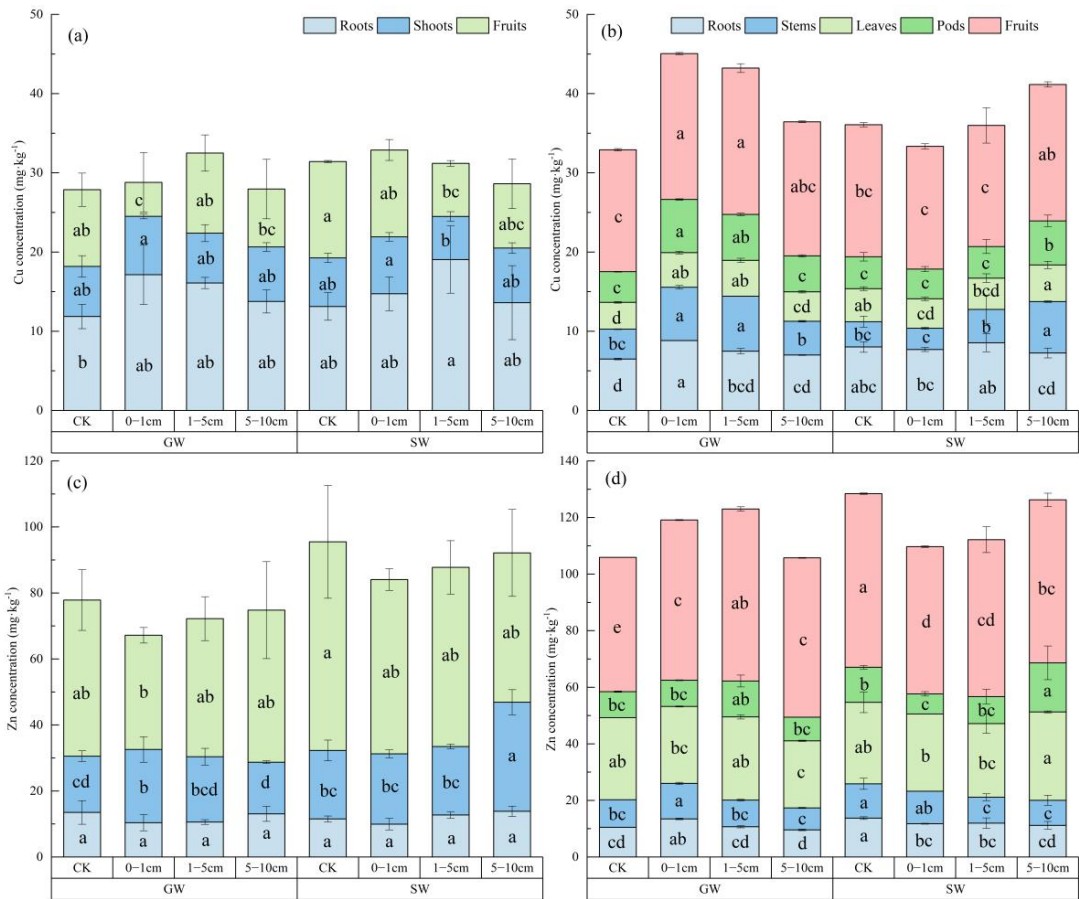

**Figure 5.** Cu concentrations in the plant tissues of maize (**a**) and soybeans (**b**) and the Zn concentrations in the plant tissues of maize (**c**) and soybeans (**d**). "CK", "0–1 cm", "1–5 cm", and "5–10 cm" represent the four straw return treatments with different sizes. GW refers to groundwater, and SW refers to swine wastewater. Different lowercase letters on the same color columns represent significant differences between treatments at $p < 0.05$.

The Cu and Zn distribution in soybean plants varied from those in maize plants (Figure 5b,d). Water significantly affected root and stem Zn as well as stem, pod, and fruit Cu; straw size significantly affected Cu and Zn in roots, stems, pods, and fruits except for fruit Cu, and their interaction affected Cu and Zn in all the organs except for stem Zn (Table 5). SW irrigation definitely enriched the Cu and Zn contents in roots as compared with GW irrigation in the CK without straw, while it did not achieve this in the straw return treatments. The Cu and Zn concentrations in the roots decreased with the increase in the size of the returning straw regardless of the irrigation water resources. When no straw was used, the concentration of Cu in the stems was not influenced by the irrigation water resources, while that of Zn rose after the application of SW. The return of straw (>1 cm) elevated the contents of the stem's Cu but reduced those of Zn relative to the CK when irrigated with SW. The Cu content in the leaves went up due to the straw return of all sizes under GW irrigation but went down as a result of straw return (<5 cm) under SW irrigation. For the Zn content in leaves, SW irrigation significantly facilitated its accumulation only in the treatments with large straws in comparison with GW irrigation. The Cu content in pods was significantly higher in the treatments using straws of 0–1 cm combined with GW than in the other treatments. The Zn concentrations in pods were unresponsive to straw return under GW irrigation but were significantly promoted by the straw addition of 5–10 cm under SW irrigation. SW irrigation led to the enrichment in fruit Cu and Zn in the no-straw control compared with that observed for GW irrigation. Straw return increased the Cu content in soybean fruits under GW irrigation irrespective of size, while the opposite

occurred under SW irrigation with straw (<5 cm). Similarly to Cu, the Zn content in fruits was increased by straw return of all sizes under GW irrigation but was lowered by straw return under SW irrigation, especially with the 0–1 cm straw treatment.

**Table 5.** Two-way ANOVA results for the heavy metals of soybean plants. Significant differences were observed at $p < 0.05$ *, $p < 0.01$ **, and $p < 0.001$ ***.

| Parts of Plants | Source of Variation | Cu | | Zn | |
|---|---|---|---|---|---|
| | | F | $p$ | F | $p$ |
| Roots | Irrigation water resource | 3.766 | 0.070 | 10.023 | 0.006 ** |
| | Straw size | 6.414 | 0.005 ** | 7.750 | 0.002 ** |
| | Interaction | 6.961 | 0.003 ** | 8.698 | 0.001 ** |
| Stems | Irrigation water resource | 21.065 | <0.001 *** | 1.655 | 0.217 |
| | Straw size | 11.479 | <0.001 *** | 14.917 | <0.001 *** |
| | Interaction | 24.080 | <0.001 *** | 3.162 | 0.053 |
| Leaves | Irrigation water resource | 0.774 | 0.392 | 1.876 | 0.190 |
| | Straw size | 2.554 | 0.092 | 1.048 | 0.398 |
| | Interaction | 11.315 | <0.001 *** | 10.101 | <0.001 *** |
| Pods | Irrigation water resource | 21.605 | <0.001 *** | 3.064 | 0.099 |
| | Straw size | 8.896 | 0.001 ** | 3.717 | 0.033 * |
| | Interaction | 22.453 | <0.001 *** | 7.892 | 0.002 ** |
| Fruits | Irrigation water resource | 10.998 | 0.004 ** | 2.966 | 0.104 |
| | Straw size | 1.930 | 0.165 | 6.359 | 0.005 ** |
| | Interaction | 10.845 | <0.001 *** | 34.974 | <0.001 *** |

*3.4. Bioaccumulation Factor*

The bioaccumulation of Cu was much easier than that of Zn, and the uptake capacity of Cu in the maize roots was higher than that in the soybean roots (Figure 6). For maize, the irrigation water resources and straw application did not alter the BFs of Cu and Zn dramatically. For soybeans, straw return accelerated the transfer of Cu from to roots when irrigated with GW with a significant acceleration in the 0–1 cm straw treatment relative to CK, while did not when irrigated with SW. Compared with the CK, the Zn BF decreased, which was ascribed to the introduction of straw of all sizes under SW irrigation. This was also true after the addition of only the 5–10 cm straw under GW irrigation.

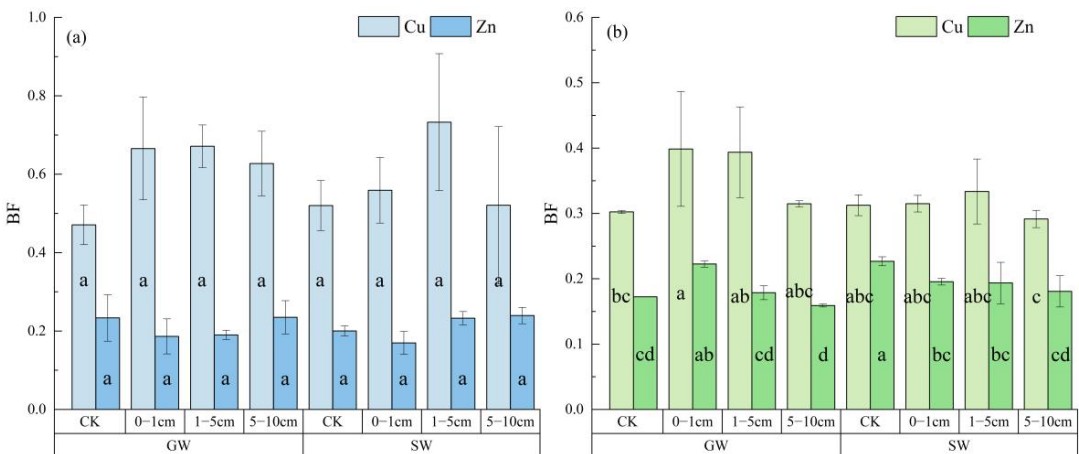

**Figure 6.** Bioaccumulation factor (BF) of heavy metals for maize (**a**) and soybeans (**b**). "CK", "0–1 cm", "1–5 cm", and "5–10 cm" represent the four straw return treatments with different sizes. GW refers to groundwater, and SW refers to swine wastewater. Different lowercase letters on the center of the light-color and dark-color columns represent significant differences in the BFs of Cu and Zn among different treatments, respectively, at $p < 0.05$.

### 3.5. Translocation Factor

In contrast to the BF, Zn in the roots was transferred to the fruits to a higher extent than Cu (Figure 7). The straw return lowered the TF of Cu in maize, which was relatively lower in the 0–1 cm straw treatment under GW irrigation and in the 1–5 cm straw treatment under SW irrigation than was observed for other treatments. SW irrigation lifted the TF of Zn when the straw size was less than 5 cm relative to GW irrigation. The TF of Zn was lessened with the straw size under SW irrigation, with the lowest value being observed for the 5–10 cm straw treatment. For soybeans, the use of straw in small sizes inhibited the transportation of Cu under GW irrigation, while the opposite was true when using large sizes of straw under SW irrigation. The TF of Zn tended to increase with the straw size when irrigated either with GW or with SW, and it was higher when the straw size was higher than 1 cm under GW irrigation.

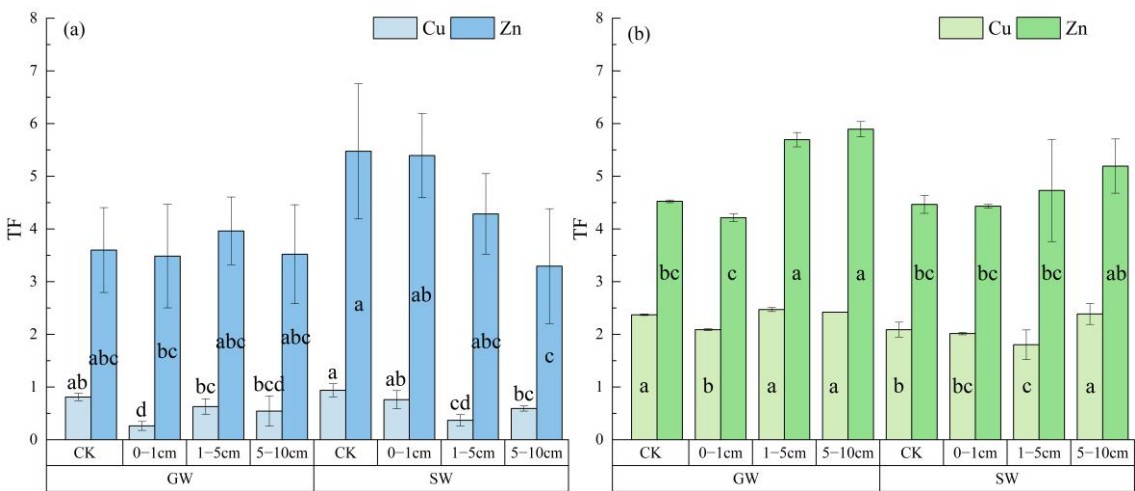

**Figure 7.** Translocation factor (TF) of heavy metals for maize (**a**) and soybeans (**b**). "CK", "0–1 cm", "1–5 cm", and "5–10 cm" represent the four straw return treatments with different sizes. GW refers to groundwater, and SW refers to swine wastewater. Different lowercase letters on the center of the light-color and dark-color columns represent significant differences in the TF of Cu and Zn among different treatments, respectively, at $p < 0.05$.

### 3.6. Associations between Rhizospheric Properties and Heavy Metal Accumulation in Plant Fruits

For maize, available P (accounting for 17.5% of the variability, pseudo-F = 4.7; $p = 0.003$), ammonium-nitrogen (accounting for 4.4% of the variability, pseudo-F = 1.2; $p = 0.318$), and nitrate-nitrogen (accounting for 2.7% of the variability, pseudo-F = 0.7; $p = 0.597$) in the rhizosphere were strongly associated with the separation of SW- and GW-irrigated treatments (Figure 3a). The first RDA axis (accounting for 17.65% of the variation) and the second RDA axis (accounting for 5.55% of the variation) accounted for 23.2% of the variation in the heavy metal distribution in soil and fruits. The increase in soil-available P and the decrease in soil ammonium showed a close association with the increase in the heavy metal content in soil and fruits. The nitrate level in the soil was positively and significantly correlated with fruit Zn and soil Cu.

There was also a clear separation of SW- and GW-irrigated treatments for soybeans along the first RDA axis (accounting for 22.02% of the variability) (Figure 3b). Except for the three variables confirmed in maize, AK (accounting for 13.5% of the variability, pseudo-F = 3.4; $p = 0.018$), OM (accounting for 13.1% of the variability, pseudo-F = 3.7; $p = 0.014$) and nitrate-nitrogen (accounting for 8.5% of the variability, pseudo-F = 2.6; $p = 0.062$) exerted a vital influence on the variation in the heavy metal content in the rhizosphere and fruits. For example, soil-available K, ammonium, and available P drove the accumulation of Cu and Zn in soil, while soil pH had the opposite effect. Soil OM was negatively associated with the enrichment of Cu and Zn in fruits, but soil nitrate worked reversely.

## 4. Discussion

This study evaluated the effects of the interaction between SW irrigation and straw return on the accumulation of heavy metals in the soil–plant system. As we hypothesized, SW irrigation and straw return could affect heavy metal mobility in the soil–plant system, and straw return inhibits the increase in heavy metals in plants caused by SW irrigation. Because of the difference in nutrient release of straw returned to the soil, the degree of inhibition observed for heavy metals in plants caused by the returning straw sizes is also different.

### 4.1. Effect of Livestock Wastewater Irrigation on Heavy Metal Mobility in the Soil–Plant System

Swine wastewater elevated the level of soil-available potassium, available phosphorus, organic nitrogen, and heavy metals in this study, which was possibly ascribed to the relatively high levels of nutrients and heavy metals in the wastewater. Wastewater irrigation can improve soil fertility, but it also increases the concentration of Cu and Zn in soils compared with groundwater irrigation [38]. Consistent with the results of Adeli et al. [39], SW irrigation reduced the pH of soybean soils in this study, which was probably due to the oxidation of organic compounds and the nitrification of ammonium in the soil [40,41]. It is generally accepted that a pH increase may lower the bioavailability of heavy metals. We also found negative correlations between soil pH and heavy metal content in plants, which have also been confirmed in previous studies [42–44]. Other studies have shown that wastewater irrigation increases the concentration of soil OM [42], while the alteration of soil OM content was not obvious in this experiment. This may be because the dilution ratio of swine wastewater was a little high before irrigation in our study in order to meet the farmland irrigation water quality standards. Despite this, both positive and negative associations between soil OM and heavy metal content in plants were established. Given that the COD concentration in wastewater was high, the increase in low-weight molecular organic acid with the increase in total OM could promote the dissolution of heavy metal hydroxides, thereby increasing the bioavailability of heavy metals and the following uptake by plants. In contrast, the macromolecular organic acids released by OM may form a complicated complex with heavy metals that is difficult to absorb, thereby reducing the bioavailability of heavy metals.

### 4.2. Effect of the Size of Returning Straw on Heavy Metal Mobility in the Soil–plant System

Straw return could reduce the accumulation of heavy metals in crops in this study. Similarly, Xu et al. [26] indicated that the application of rice and wheat straw to heavy metal-polluted soil significantly reduced shoot Cd and Pb concentrations and thus reduced Cd and Pb accumulation in maize. Moreover, the smaller the size of the straw returned to the soil the lower the heavy metal content accumulated in the plant organs was, and this effect was more obvious in Zn (Figure 5). This may be associated with the differences in the adsorption capacity of the different sizes of straw to the heavy metals in the soil due to differences in the specific surface area. Guo et al. [45] also proved that the addition of organic materials, such as biochar, wheat straw, and cow manure compost, could improve the adsorption of heavy metals in soil. The decomposition process and nutrient release of straw with smaller sizes were faster and promoted plant growth, thus reducing the content of heavy metals in plants due to the enhanced biomass (bio-dilution effect). The components of straw, hemicellulose, and lignin, and their degradation products might also affect the adsorption [46–48], forms, and bioavailability of heavy metals.

### 4.3. Interaction between Wastewater Application and Straw Return

The reducing effect on the heavy metal content in plants under SW irrigation was more noteworthy than that under GW irrigation, indicating an interaction between SW and straw. The carbon-to-nitrogen ratio of straw is usually high [49], and its decomposition needs sufficient nitrogen, which was supplemented exactly by the introduction of livestock wastewater. Therefore, livestock wastewater accelerated the decomposition of straw re-



sulting in the reduced bioavailability of heavy metals after their complexation with the degradation products of straw. Furthermore, livestock wastewater irrigation lowered soil pH and introduced other nutrients to the soil (Figure 2), which may provide energy for soil microorganisms and improve their structure. Incidentally, the abundance of soil bacteria and fungi [50] that can decompose straw was also promoted. Under conditions with rich nutrients, the formation of biofilm on the surface of the straw also increases the adsorption of heavy metals onto the straw. The inhibitory effect of the small-sized (<1 cm) straw on the heavy metal accumulation in crops under irrigation with farm wastewater was more pronounced. This may be due to the different decomposition processes of straw in soil after returning to the field; the small-sized straw released macromolecular organic matter more quickly when irrigated with farm wastewater, which reduced the bioavailability of heavy metals [51].

Based on the results, the use of diluted swine wastewater for irrigation had less effect on the soil's basic properties and heavy metals, but it increased the uptake of heavy metals by crops. Although the swine wastewater was diluted, the concentration of soluble heavy metals contained in it was still high relative to GW, which could be absorbed easily by crops [52]. In addition, the release of organic matter by straw return changed the speciation of heavy metals and reduced their bioavailability [27], but it did not significantly affect the total amount of heavy metals in the irrigated soils.

In addition, heavy metals can also be introduced into the soil by the application of phosphate fertilizers (e.g., diammonium phosphate and superphosphate) [53] and other fertilizers. Fertilizers are likely to alter soil properties such as pH and surface charge, and they could react with heavy metals in soil, resulting in changes in the forms and therefore the bioavailability of heavy metals [54]. Phosphate addition has been found to increase the cation exchange capacity of soil, enhancing metal sorption by soil [55]. In this experiment, we added the same amount of compound fertilizer to each pot, so the effects of the compound fertilizer on the soil properties and heavy metals in different treatments were not compared.

### 4.4. Performance Differences between Cu and Zn

The accumulation of Zn in crops was higher than that of Cu in this study, which is consistent with the heavy metals in the wheat grains collected in Kunshan City, China [56]. This is acceptable since the Zn content in SW and soil in our experiment was higher than that of Cu, and both Cu and Zn are essential trace elements in crop growth. However, the correlation between Cu in soil and Cu in fruits was lower than that between Zn in soil and Zn in fruits (Figure 3), that is to say, Zn is more easily accumulated in crops than Cu. Variations in other soil properties, the nature of the two metals, and their different nutritional functions in plants can also contribute to the distinctions in heavy metal accumulation in crops. Compared with Zn, Cu is more easily adsorbed by soil. The two metals respond differently to the pH change. Here, only Zn in fruit is significantly negatively correlated with soil pH. Although Cu and Zn co-exist in superoxide dismutase to protect chloroplasts from superoxide free radicals, Cu can promote amino acid activation and protein synthesis in the process of protein formation, and Zn is also involved in the synthesis of the growth hormone (indole acetic acid) in plants.

### 4.5. Performance Differences between Maize and Soybean

Under these experimental conditions, the plantation of maize and soybean exerted different influences on the soil properties and heavy metal contents in plants. Maize is a C4 crop with a whisker root system, while soybean is a C3 crop with a straight root system with rich porosity and high absorption efficiency, which led to the differences in soil water and nutrient utilization between the two crops—the content of AP in the maize soil was slightly higher than that in the soybean soil (Figures 1 and 2). The mucous and soluble exudates secreted by maize roots have a strong ability to form complexes with Cu [57–59], which promoted the solubility of metal and thus increased the Cu BF (Figure 6) compared

with soybeans. Meanwhile, the growth of roots, especially with the optimized structure, helps roots to absorb more nutrients and water [60,61]. The roots of soybeans were mainly concentrated in the surface layer (0–20 cm), while those of maize were widely distributed (0–40 cm) [62]. Although the organic acids secreted by roots decreased the metal sorption by the soil particles [63], the transport of most heavy metals in maize was weak, leading to less accumulation of heavy metals in maize shoots and fruits [64]. Consequently, heavy metal concentrations in maize roots were higher than those in soybean roots (Figure 5), but the Cu TF of soybeans was higher than that of maize (Figure 7). This indicated that the heavy metals absorbed in the surface layer by soybean roots were transferred more to the soybean fruits, while this was not true for the maize roots, although they did accumulate more heavy metals.

In addition, as a leguminous crop, soybean plants may introduce nitrogen from the atmosphere, which can affect the soil properties and heavy metal absorption [65]. Certain studies have proven that a high supply of nitrogen inhibited nodule growth and nitrogenase activity [66,67]. The nitrate-nitrogen content in the soils of our experiment was high, and it can be assumed that soybean plants did not undergo nitrogen fixation in this experiment.

## 5. Conclusions

The effects of straw returns with different sizes under swine wastewater irrigation on the accumulation of heavy metals were investigated in a pot experiment. We found that the combined application of swine wastewater irrigation and straw return did not significantly alter soil pH, organic matter, or other properties, nor did it aggravate the accumulation of heavy metals in the soil. Irrigation with swine wastewater increased the content of heavy metals in plants; however, straw return reduced this increase, especially when the straw size was small. According to the existing data, the concentration of Cu and Zn in soils and plants were within the acceptable limits. Therefore, livestock wastewater can be used as an alternative irrigation water resource, and the accompanied accumulation of heavy metals in crops can be reduced with straw return. In actual production, the dynamics of the properties in soil irrigated with livestock wastewater including heavy metals and other emerging contaminates should be monitored to ensure soil health. Meanwhile, returning straw of a smaller size to the field has the potential to reduce the accumulation of heavy metals in crops.

**Author Contributions:** Conceptualization, S.L. (Siyi Li), Y.L. and Z.L.; data curation, S.L. (Siyi Li), Z.T., S.L. (Shengshu Li), R.K. and C.H.; formal analysis, S.L. (Siyi Li) and Y.L.; funding acquisition, Y.L. and Z.L.; supervision, Y.L. and Z.L.; methodology, S.L. (Siyi Li) and X.F.; writing—original draft preparation, S.L. (Siyi Li); writing—review and editing, Y.L. and Z.L. All authors contributed critically to the drafts and gave final approval for publication. All authors have read and agreed to the published version of the manuscript.

**Funding:** This work was funded by the National Key Research and Development Program of China (2021YFD1700900), the Central Public-Interest Scientific Institution Basal Research Fund (FIRI2022-04 and Y2022LM29), the National Natural Science Foundation of China (41701265), the Talent Cultivation Program of the Chinese Academy of Agricultural Sciences (NKYCQN-2021-028), and the Agricultural Science and Technology Innovation Program (ASTIP) of the Chinese Academy of Agricultural Sciences.

**Institutional Review Board Statement:** Not applicable.

**Data Availability Statement:** The data presented in this study are available on request from the corresponding author.

**Conflicts of Interest:** The authors declare no conflicts of interest.

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
