# Peer review of "Influence of Swine Wastewater Irrigation and Straw Return on the Accumulation of Selected Metallic Elements in Soil and Plants"

_agriculture, doi:10.3390/agriculture14020317_

Round 1

Reviewer 1 Report

Comments and Suggestions for Authors

Please see attached

Comments on the Quality of English Language

The language is a bit complicated in some places. Needs checking and simplification.

Author Response

Dear Reviewers/Editor,

We are grateful for the time and effort the reviewers have taken to help us improve the manuscript. Our responses to Reviewer #1 are marked in red in the revised manuscript.

Reviewer #1:

The suggestions for your review article are listed below. 

  1. Title: Straw Return Inhibits Heavy Metal Accumulation in Crops Under Swine Wastewater Irrigation

Comment: The title sounds like a research hypothesis. It should be modified accordingly. The hypothesis can, of course, be put in the work, but in the Methodology chapter or in the Introduction before the purpose of the work.

Proposition of title: Influence of swine wastewater irrigation and straw return on the accumulation of selected metallic elements in soil and plants.

Reply: We revised as suggested. Line 2-4 and 100-101.

  1. Straw is not the only source of organic substances in the soil, and it is the organic substance that affects the bioavailability of metallic elements. I think that in the Introduction the authors should also raise the problem of the level of organic substances in soils in China or other countries of the world, for comparison. In many regions of the world, the level of organic matter in the soil has declined significantly over the last few decades. This is influenced by climate change, lower water availability and faster mineralization, and the use of liquid natural fertilizers, which migrate much faster in the soil, but also contain less organic substances. Intensive farming and agricultural practices also contribute to the reduction of organic matter levels. In addition, we should also remember about soil organisms, which are very important from the point of view of agricultural crops. The influence of metallic elements on these organisms is also an important aspect. It is worth raising these problems in the Introduction in the environmental and climate context.

Reply: Thank you for your suggestion. We included the above aspects in the introduction. Line 27-35 and 89-93.

  1. Table 2 and 3. Please explain the abbreviations a, b, c, d.

Reply: We changed the tables into figures for better understanding, and reexplained the abbreviations. Line 202-207 and 220-226.

  1. Has the chemical composition of straw been tested? If not, it may be worth providing at least the content of NPK and the tested metallic elements in wheat straw collected in China from the literature. Straw can influence the nitrogen transformation processes in the soil, but also increase potassium concentration.

Reply: The chemical composition of straw was added. Line 124-127.

  1. Please explain what was the rooting depth of corn and soybeans? This also affects the absorption of elements from the soil.

Reply: We have added the contents in the discussion. Line 473-483.

  1. How long did the fermentation process of pig sewage last before it was used? How much organic matter was there in pig sewage in its original state and after dilution?

Reply: The fermentation process of pig sewage lasted about 30 days before it was used. For the organic matter, we only measured the chemical oxygen demand (COD) in swine wastewater to ensure that the swine wastewater met the quality requirements of irrigation water. Line 117-118.

  1. What about nitrogen in soybean cultivation? It is a legume that introduces a certain amount of nitrogen into the soil from the atmosphere. Did the authors wonder what effect this might have on pH, nitrogen metabolism or the absorption of the tested metallic elements?

Reply: Some studies have proved that high nitrogen supply inhibits nodule growth and nitrogenase activity of soybean. The nitrate nitrogen content in the irrigated soils in this experiment was high and we did not notice the nodules on the roots, so it can be assumed that soybean did not undergo nitrogen fixation in this experiment. However, this is a meaningful suggestion, and we will fully consider the nitrogen fixation of soybeans and its effects on soil properties and heavy metal absorption in future experiments. We also added the contents in the discussion. Line 484-488.

  1. Were the concentrations of metallic elements in the soil within acceptable limits after irrigation? This is important information in an environmental context.

Reply: The total amount of Cu and Zn in the irrigated soil were lower than 28 mg/kg and 61 mg/kg which did not exceed the requirement of risk control standard for soil contamination of agricultural land. Line 266-270.

  1. What about leaching surplus metallic elements from the soil? Have soil leachates and element migration been tested? This is an important element in the context of groundwater and groundwater contamination.

Reply: The soil moisture of the pots was maintained at 60%–70% of the water holding capacity. The pots were irrigated every 3–4 days with 1 L of water and no water came out of the pot bottom after each watering. Soil leaching was not involved in this experiment so we didn’t test the soil leachates and element migration. Line 138-141.

  1. Conclusions: The conclusions are a bit too short and too general for such a broad study, and they are a repetition of the results. The authors should go a step further in their conclusions. Having obtained the results, the authors should consider how they can be used in practice. Is it worth irrigating with pig sewage or not? If it is worth it, under what conditions, in what dilutions, in what regions. There are soils with an increased content of metallic elements, or those where the concentrations of the elements exceed the permitted standards. What about the use of pig sewage irrigation on slopes? Are there any environmental risks that result from such irrigation? How can farmers approach this type of practice?

Reply: Thank you for the suggestion. We have rewritten the conclusion. Line 490-503.

  1. The language is a bit complicated in some places. Needs checking and simplification.

Reply: Thank you for the suggestion. We revised some sentences in Line 74-77, 246-247 and 450-451.

Reviewer 2 Report

Comments and Suggestions for Authors

Line 12 maize or soybean - you don't know?
Line 41-43 any references? Wastewater always contains pollutants - even treated ones
Introduction:
The work focuses on the behaviour of Cu and Zn, please expand the standards for these specific elements in the introduction. The description of the others, e.g. Pb, Co, Cd - obviously important from an environmental point of view - is of negligible importance for this particular work.
Line 91-92 please explain wht for pH and EC water-soil ratio was 5:1 as it's rather 2,5:1 for pH and 2:1 for EC.
Line 94-95 what was the ratio for water:wastewater?
Line 98 SW:GW 1:40?
Line 106 how much is it in t/ha?
Line 106-107 don't you think that 30g of fertilizer had bigger effect on plant production that 8g of straw? How much soil was in each pot? There is no inforamtion on how was it applied.
Line 109 please specify how many pots has maize and how many soybean. For now i understand that you plant seeds randomly.
Line 111 how was irrigation applied?
Line 112 what about SW?
Line 176 what is CK? control?
Table 2/3 please reconsider data presentation as in this version it's extremly hard to understand. what is a, .01a for?
Why do you present data separatelly for GW and SW if swine wastewater was diluted at a ratio of 1:40 every time before use (line 98)?
Line 335-338 all depends on soil profile and its layers.
Line 339-408 In my opinion, the results obtained are discussed rather superficially and require a slightly deeper look. The work involved fertilization with organic matter and wastewater - which, according to the methodology, was diluted, but this is not visible in the results. Mineral fertilization was completely omitted (30g per pot).
The conclusions drawn are very sketchy and require rewriting.

Author Response

Dear Reviewers/Editor,

We are grateful for the time and effort the reviewers have taken to help us improve the manuscript. Our responses to Reviewer #2 are marked in blue in the revised manuscript.

Reviewer #2:

The suggestions for your review article are listed below.

  1. Line 12 maize or soybean - you don't know?

Reply: We are sorry that this sentence caused ambiguity, and we have rephrased this sentence. Line 13-15.

  1. Line 41-43 any references? Wastewater always contains pollutants - even treated ones

Appropriate references have been added.

Reply: We have added the references. Line 51.

  1. Introduction: The work focuses on the behavior of Cu and Zn, please expand the standards for these specific elements in the introduction. The description of the others, e.g. Pb, Co, Cd - obviously important from an environmental point of view - is of negligible importance for this particular work.

Reply: We have adjusted this section in the introduction. Line 62-71.

  1. Line 91-92 please explain why for pH and EC water-soil ratio was 5:1 as it's rather 2,5:1 for pH and 2:1 for EC.

Reply: The water-soil ratio 5:1 is commonly used for the pH determination of soil with high salt content (Lu Rukun. Analysis Method of Agricultural Chemistry in Soil. Beijing: Agriculture and Science Press, 1999. (in Chinese)). In this study, the soil salt content is high, so we used the water-soil ratio 5:1. According to the standard determination method of soil conductivity in China (HJ 802-2016, https://www.mee.gov.cn/ywgz/fgbz/bz/bzwb/jcffbz/201606/t20160630_356524.shtml), the water-soil ratio of EC was unified as 5:1.

  1. Line 94-95 what was the ratio for water: wastewater?

Reply: Two irrigation water resources, groundwater and swine wastewater, were used in this experiment, and the amount of irrigation water was kept the same for all treatments. Line 136-138.

  1. Line 98 SW: GW 1:40?

Reply: No, to minimize the disturbance of water quality by the dilution process, we diluted swine wastewater 40 times using deionized water to conform to the standard for irrigation water quality. Line 118-122.

  1. Line 106 how much is it in t/ha?

Reply: 80 g straw per pot (30 kg of soil, bulk density of 1.35 g/cm3), equal to 7.2 t/ha, was applied in this study. Line 131.

  1. Line 106-107 don't you think that 30g of fertilizer had bigger effect on plant production that 8g of straw? How much soil was in each pot? There is no information on how was it applied.

Reply: I agree. The nutrient provided by 30 g of compound fertilizer is greater than that of 80 g of straw, therefore the effect on plant production is also bigger. There was 30 kg of the soil in each pot, which was provided in Line 109 in the original manuscript and Line 134 in the revised version. According to the actual straw production situation in the field, we added 80 g of straw to each pot. Line 131-134.

  1. Line 109 please specify how many pots has maize and how many soybean. For now I understand that you plant seeds randomly.

Reply: There are 24 pots of maize plants and 24 pots of soybean plants. Four seeds per pot were planted at the beginning, and after two weeks of germination, the seedlings per pot was reduced to 1. Line 134-136.

  1. Line 111 how was irrigation applied?

Reply: The soil moisture of the pots was maintained at 60%–70% of the water holding capacity. And the pots were irrigated every 3–4 days. Water of 1 L were measured using a beaker with a scale and then evenly pour into each pot each time. Line 136-141.

  1. Line 112 what about SW?

Reply: Prior to thinning, all treatments were irrigated with groundwater. After thinning, the pots were irrigated with GW or SW according to the treatment design, and the irrigation amount was same across all the treatments. Line 137-138.

  1. Line 176 what is CK? control?

Reply: CK was the treatment without straw. Line 124.

  1. Table 2/3 please reconsider data presentation as in this version it's extremely hard to understand. what is a, .01a for?

Reply: Due to the table template problem, the data originally in one row was broken into two rows. To figure out this, we have changed the tables into figures. Line 202-207 and 220-226.

  1. Why do you present data separately for GW and SW if swine wastewater was diluted at a ratio of 1:40 every time before use (line 98)?

Reply: As both groundwater and swine wastewater are variables in this experiment, their basic properties are listed. Line 121-122.

  1. Line 335-338 all depends on soil profile and its layers.

Reply: We have rewritten this part. Line 382-384.

  1. Line 339-408 In my opinion, the results obtained are discussed rather superficially and require a slightly deeper look. The work involved fertilization with organic matter and wastewater - which, according to the methodology, was diluted, but this is not visible in the results. Mineral fertilization was completely omitted (30g per pot).

Reply: Fertilizers from different sources had different effects on soil properties and heavy metals. We supplemented the discussion. Line 428-448.

  1. The conclusions drawn are very sketchy and require rewriting

Reply: Thank you for the suggestion. We have rewritten the conclusion. Line 490-503.

Round 2

Reviewer 2 Report

Comments and Suggestions for Authors

The authors have significantly improved the value of the paper, including an improved description of the methodology used to conduct the analysis. In its present form, the paper is suitable for publication.